

# Evaluating the effectiveness of decomposed Halstead Metrics in software fault prediction

Bilal Khan and Aamer Nadeem

Capital University of Science & Technology, Islamabad, Pakistan

## ABSTRACT

The occurrence of faults in software systems represents an inevitable predicament. Testing is the most common means to detect such faults; however, exhaustive testing is not feasible for any nontrivial system. Software fault prediction (SFP), which identifies software components that are more prone to errors, seeks to supplement the testing process. Thus, testing efforts can be focused on such modules. Various approaches exist for SFP, with machine learning (ML) emerging as the prevailing methodology. ML-based SFP relies on a wide range of metrics, ranging from file-level and class-level to method-level and even line-level metrics. More granularized metrics are expected to possess a higher degree of micro-level coverage of the code. The Halstead metric suite offers coverage at the line level and has been extensively employed across diverse domains such as fault prediction, quality assessment, and similarity approximation for the past three decades. In this article, we propose to decompose Halstead base metrics and evaluate their fault prediction capability. The Halstead base metrics consist of operators and operands. In the context of the Java language, we partition operators into five distinct categories, *i.e.*, assignment operators, arithmetic operators, logical operators, relational operators, and all other types of operators. Similarly, operands are classified into two classes: constants and variables. For the purpose of empirical evaluation, two experiments were designed. In the first experiment, the Halstead base metrics were used along with McCabe, Lines of Code (LoC), and Halstead-derived metrics as predictors. In the second experiment, decomposed Halstead base metrics were used along with McCabe, LoC, and Halstead-derived metrics. Five public datasets were selected for the experiments. The ML classifiers used included logistic regression, naïve Bayes, decision tree, multilayer perceptron, random forest, and support vector machines. The ML classifiers' effectiveness was assessed through metrics such as accuracy, F-measure, and AUC. Accuracy saw an enhancement from 0.82 to 0.97, while F-measure exhibited improvement from 0.81 to 0.99. Correspondingly, the AUC value advanced from 0.79 to 0.99. These findings highlight the superior performance of decomposed Halstead metrics, as opposed to the original Halstead base metrics, in predicting faults across all datasets.

Corresponding author
Bilal Khan, drbilal345@gmail.com

# INTRODUCTION

In the domain of software systems, the occurrence of faults is an unavoidable predicament. The manifestation of software faults has the potential to generate significant losses and catastrophic consequences. These faults are intimately linked with the security, reliability, and maintainability of the system. Testing is a crucial activity aimed at detecting the presence of faults. Nonetheless, within software projects, it is an arduous and costly endeavor for testers to find all the software faults. As a matter of fact, it has been reported that testing and quality assurance consume roughly 35% of the total development cost (*Capgemini Group, 2015*). Furthermore, with the progressive increase in the complexity of software programs, characterized by a vast number of lines of code, the occurrence of software faults becomes inevitable (*Zakari & Lee, 2019*). It has been ascertained that faults in software systems are not uniformly distributed, as reported by *Sherer (1995)*. Certain modules in a software product are more prone to carrying faults than others. *Chappelly et al. (2017)* found that faults were present in only 42% of software modules in a software system. Moreover, a separate study has indicated that approximately 70% of faults are discovered by testing only 6% of the software modules (*Abaei & Selamat, 2014*). These results are consistent with multiple studies conducted by *Weyuker, Ostrand & Bell (2008)* and *Watanabe, Kaiya & Kaijiri (2008)*. SFP is a formal process that involves the identification of fault-prone *fp* module(s) within a software system or the estimation of the expected number of faults in a particular software module (*Rathore & Kumar, 2017b*; *Sandeep & Santosh, 2018*). The ability to detect faulty modules or estimate the number of faults in a module in a timely manner is particularly beneficial for critical and strategic software systems as it helps to reduce testing costs and improve the quality of the software (*Sandeep & Santosh, 2018*). Furthermore, SFP facilitates the targeting of testing efforts towards *fp* modules, ultimately leading to improved testing efficacy. The prediction of the number of faults is also valuable as it provides a criterion for determining the sufficiency of testing. Additionally, SFP contributes to the optimal utilization of resources in the software testing process (*Seliya & Khoshgoftaar, 2007*). Typically, there exist three distinct methodologies that are employed in SFP.

1. **Experts' opinion:** The experts' opinion on SFP emphasizes the necessity of a comprehensive approach to data collection and analysis, which includes historical fault data, code metrics, and information about the software development process. They advocate for the use of software metrics to derive meaningful patterns and relationships from the collected data, enabling the identification of potential *fp* areas in software systems. Additionally, experts consider SFP as an integral part of a broader software quality assurance process. They recommend the integration of fault prediction with other quality assurance activities such as code reviews, testing, and software maintenance, to ensure a holistic approach to software quality.

2. **Statistical modeling:** One of the key approaches to SFP is statistical modeling. This refers to the use of statistical techniques and models to analyze software data and make predictions about the occurrence of faults or defects in software systems. Statistical modeling involves employing various approaches, such as regression analysis and

time series analysis, to identify patterns, relationships, and trends in software metrics or other relevant data. To build predictive models that can estimate the likelihood of future software faults, statistical models are trained on historical software data. This data includes software metrics, fault records, and other contextual information. By learning patterns from this data, predictive models can be created to forecast the occurrence of future software faults.

3. **Machine learning:** The utilization of machine learning algorithms for SFP is an approach in which historical data related to software development and testing is analyzed to identify patterns and relationships, and predictive models are constructed to forecast potential faults. This approach involves applying machine learning techniques to predict software faults or defects in a system.

Among them, machine learning (ML) is the most dominant method in use due to its high accuracy, as noted in a previous study (*Catal & Diri, 2009*). The main requirement in machine learning based SFP is to train a fault prediction model using a labeled dataset. The dataset consists of features (software metrics) of software modules and a label (*i.e.,* faulty or non-faulty). These metrics can be categorized as product metrics or process metrics. Process metrics relate to the software development process. Examples of such metrics include programmer experience level, defects found in reviews, and the amount of time spent in reviews on a module. On the other hand, product metrics describe the characteristics of the software product, such as its size, complexity, design features, performance, and quality level. According to studies reported by *Catal & Diri (2009)*, product metrics are used in 79% of the SFP studies. The coverage of product metrics can be at the class level, method level, or file level. Class-level metrics are used in object-oriented programming due to the fact that the class concept represents the abstract characteristic of an object in an object-oriented paradigm. Among the various metric suites used in the object-oriented paradigm, the CK metric suite is the most frequently used, as reported in several studies (*Radjenović et al., 2013*; *Catal & Diri, 2009*; *Beecham et al., 2010*; *Gondra, 2008*; *Malhotra, 2015*; *Chappelly et al., 2017*; *Nam et al., 2017*; *Li, Jing & Zhu, 2018*). Other metric suites specific to the object-oriented paradigm, such as MOOD (*Abreu & Carapuça, 1994*; *Lorenz & Kidd, 1994*), are also available. Method-level metrics, such as Fan-in, Fan-out, and McCabe, capture the method-level aspects of a software system. Furthermore, certain metrics have the potential to compute line-level aspects of a software system, with the Halstead metrics suite being one such metric. According to research, method-level metrics are the most widely used (60%), followed by class-level metrics, which account for about 24% (*Catal & Diri, 2009*). A comprehensive discussion of software metrics is available in *Ghani (2014)*; *Fenton & Bieman (2014)*; *Abbad-Andaloussi (2023)*. Moreover, software metrics have a strong relation with cognitive complexity, which interns leads to software faults (*Abbad-Andaloussi, 2023*). It is worth noting that a more granular metric, such as the Halstead metrics suite, can provide greater coverage of the code at a micro level. The Halstead base metrics are comprised of four fundamental metrics: total operators (N1), total operands (N2), unique operators(n1), and unique operands (n2). According to Halstead's definition as outlined in *Halstead (1972)*, ''operands'' refer to variables and constants, while ''operators'' encompass all symbols, combinations of symbols, punctuation marks,

arithmetic, keywords, special symbols, and function names. These base metrics are further composed of operators and operands. In addition to these base metrics, seven derived metrics have been developed, including length, level, difficulty, volume, effort, program time, and error estimation.

Feature decomposition in machine learning pertains to the systematic procedure of decomposing intricate input features into more straightforward and informative representations. This endeavor encompasses the dissection of the features into a reduced feature set (*Caiafa et al., 2020*). Through the extraction of actionable elements from a column, novel features are generated, enabling machine learning (ML) algorithms to achieve a deeper understanding of the data. Consequently, this enhancement in comprehension leads to an improvement in model performance by unearthing latent information (*Jiawei, Jian & Micheline, 2000*; *Caiafa et al., 2020*).

In this article, we present our proposal to decompose the Halstead base metrics and assess their predictive performance in software fault prediction (SFP). Specifically, we decompose the metric known as "number of operators" into its constituent types, such as arithmetic operators, relational operators, logical operators, and so on. Similarly, the metric "number of operands" is decomposed into various types, including variables, constants, and others. Since datasets containing decomposed Halstead metrics were not readily accessible, we opted to utilize five datasets for which the corresponding source code was available. By parsing the source code using our customized parser, we were able to extract the decomposed Halstead metrics from these datasets.

In recent years, the research community in the field of SFP has been predominantly concentrated on the classification of *fp* and *nfp* modules, as highlighted in various studies (*Rathore & Kumar, 2017a*; *Catal, 2011*). In our investigation, we employed machine learning algorithms based on classification, namely logistic regression, naïve Bayes, decision tree, multilayer perceptron, random forest, and support vector machines.

The structure of the article is organized as follows. Firstly, the "Research Questions" section provides a concise overview of the research direction and objectives. It outlines the specific questions that the study aims to address. The subsequent section, "Related Work", briefly examines previous studies that have utilized the Halstead metrics suite in SFP. This section provides context and highlights the existing literature and research in this area. Moving forward, the "Decomposition of Halstead Base Metrics" section delves into the concept of decomposition and its significance in relation to Halstead base metrics. It expands on how decomposing these metrics can enhance understanding and analysis. In the "Methodology" section, the article focuses on describing the specific methodology employed for the empirical evaluation of Decomposed Halstead base metrics. This section highlights the approach taken in the research process. The "Experimentation" section is dedicated to detailing the experiments conducted as part of the study. It explains the experimental setup, variables, and procedures employed to gather data and insights. Next, the "Results and Discussion" section presents the findings derived from the materials and methods discussed earlier. This section reports the results computed during the research and provides a comprehensive discussion and analysis of these findings. To ensure the validity of the results, the "Threat to Validity" section assesses potential limitations and

threats to the validity of the study's findings. It acknowledges any potential biases or constraints that could affect the interpretation of the results. Finally, the "Conclusion and Future Direction" section concludes the article by summarizing the key findings and implications. It also suggests potential avenues for future research, offering possible directions for further exploration and investigation in the field.

## RESEARCH QUESTIONS

**RQ 1:** How can the Halstead base metrics be decomposed?
**Rationale:** This research question seeks to investigate the process and methodology involved in decomposing the Halstead base metrics. It aims to explore the potential approaches and techniques that can be employed to break down the Halstead base metrics into its constituent components or sub-metrics. By addressing this question, the study aims to contribute to the understanding of the decomposition process and its implications for improving the efficiency and effectiveness of the Halstead metrics suite in software analysis and evaluation. The section 'Decomposition of Halstead base metrics' is dedicated to answering this question.

**RQ 2:** What is the impact of decomposed Halstead base metrics in SFP?
**Rationale:** The research question aims to investigate the impact of the decomposed Halstead base metrics in SFP. It encourages us to analyze and compare the results obtained using the decomposed Halstead base metrics against the traditional Halstead base metrics, identify suitable performance measures, consider the influence of different datasets, and explore the applicability of the decomposed Halstead base metrics in various software engineering tasks. The section 'Results and Discussion' is dedicated to answering this question.

## RELATED WORK

Our research endeavors to conduct a comprehensive analysis of the decomposed Halstead base metrics with the purpose of evaluating its efficacy in the realm of SFP. To accomplish this, we have undertaken a literature review with two primary objectives. Firstly, we have sought to investigate the prevailing trends in SFP, particularly in relation to Machine Learning (ML) algorithms and datasets. This exploration has aided us in the selection of appropriate datasets and ML algorithms for our experimental investigations. Secondly, we have aimed to identify the commonly employed software metrics in conjunction with the Halstead metrics suite for SFP. This objective holds significant importance as SFP typically involves the integration of diverse software metrics with ML techniques for the prediction of software faults. In order to achieve our first objective, we have conducted a review of systematic literature reviews (SLRs) to promptly identify the current trends in SFP, with a specific focus on datasets and ML algorithms. For our second objective, we have examined studies in SFP that have exclusively utilized the Halstead metric suite within their metric set for the prediction of software faults.

*Catal & Diri (2009)* emphasizes several key findings in SFP research, including the widespread utilization of method-level metrics, the growing prevalence of public datasets, and the increased adoption of machine learning techniques. These observations serve

as important insights for advancing the field and enhancing the precision and efficacy of SFP models. By recognizing and implementing these recommendations, researchers and practitioners can contribute to the ongoing development and refinement of SFP methodologies.

*Catal (2011)* presents a comprehensive survey of the software engineering literature on SFP, covering both machine learning-based and statistical-based approaches. The survey findings indicate that a significant proportion of the studies examined in this survey concentrate on method-level metrics, with machine learning techniques being the primary approach employed for constructing prediction models. Notably, the study suggests that naïve Bayes emerges as a robust machine learning algorithm suitable for supervised SFP.

*Hall et al. (2012)* highlights the presence of exemplary fault prediction studies while emphasizing the existence of unresolved inquiries concerning the development of efficient fault prediction models for software systems. It asserts the necessity for additional studies that conform to reliable methodologies and consistently document contextual details and methodologies. The accumulation of a larger body of such studies would facilitate meta-analysis, provide practitioners with the confidence to adeptly choose and implement models in their systems, and ultimately augment the influence of fault prediction on the quality and cost of industrial software systems.

*Radjenović et al. (2013)* contributes significant insights regarding the feasibility of software measures for defect prediction. It presents recommendations to select metrics and underscores the crucial role of realistic validation and industrial relevance in shaping future research endeavors.

*Malhotra (2015)* review that focuses on evaluating the performance of machine learning (ML) techniques in Software SFP. The review involves analyzing the quality of 64 primary studies conducted between 1991 and 2013. The characteristics of these studies, including metrics reduction techniques, metrics used, data sets, and performance measures, are summarized. The performance of ML techniques in SFP is assessed by comparing them to models predicted using logistic regression. Furthermore, the performance of ML techniques is analyzed in comparison to other ML approaches.

*Wahono (2015)* analyzes 71 studies published between 2000 and 2013 to understand trends, datasets, methods, and frameworks used in software defect prediction. The research primarily focuses on estimation, association, classification, clustering, and dataset analysis. Classification methods dominate the studies, accounting for 77.46%, followed by estimation methods at 14.08%, and clustering/association methods at 1.41%. Public datasets are utilized in 64.79% of the studies, while private datasets are used in 35.21%. The review identifies seven frequently employed methods: logistic regression, naïve Bayes, k-nearest neighbor, neural network, decision tree, support vector machine, and random forest.

*Rathore & Kumar (2017b)* provides an examination of SFP through a comprehensive analysis of the existing literature. The review encompasses various aspects including software metrics, fault prediction techniques, concerns related to data quality, and evaluation measures for performance. By exploring these domains, the review sheds light on the challenges and methodological issues that are inherent in this field. Existing studies predominantly concentrate on object-oriented (OO) metrics and process metrics, and

they primarily utilize publicly available data. Statistical techniques, particularly binary class classification, are widely employed in these studies. A considerable amount of attention has been given to tackling issues such as high data dimensionality and class imbalance quality. Evaluation metrics like accuracy, precision, and recall are commonly used in assessing the performance of these fault prediction techniques.

*Caulo (2019)* introduces a comprehensive taxonomy of metrics for SFP. The taxonomy comprises a total of 526 metrics employed in research papers published from 1991 to 2017. The article emphasizes the significance of evaluating the efficacy of each metric in SFP. Additionally, the author proposes to categorize the identified metrics based on their co-linearity, thereby facilitating the exploration of relationships between different metrics and their collective influence on SFP.

*Pandey, Mishra & Tripathi (2021)* presents a comprehensive analysis of machine learning-based methods for SFP. Its primary objective is to explore and summarize the current state of the field by examining a diverse range of ML techniques and approaches utilized in SFP. The survey underscores the importance of leveraging machine learning for fault prediction and acknowledges its potential to improve software quality and reliability. It delves into various machine learning algorithms, including decision trees, support vector machines (SVM), neural networks, and ensemble methods, and investigates their specific applications in the context of fault prediction.

*Pachouly et al. (2022)* provides a comprehensive overview of software defect prediction using artificial intelligence (AI). It covers four key aspects: datasets, data validation methods, approaches, and tools. The review emphasizes the importance of high-quality datasets and explores different validation methods to ensure accurate and reliable data. It discusses various AI techniques and algorithms used in defect prediction, highlighting their strengths and limitations. Additionally, it identifies and examines tools and frameworks that aid in implementing and evaluating AI models for defect prediction.

A number of studies in the field of SFP have provided significant insights and recommendations, contributing to the advancement of this field. These studies have observed the widespread utilization of method-level metrics, the increasing availability of public datasets, and the growing adoption of machine learning techniques. They have also highlighted the concentration on method-level metrics and the effectiveness of algorithms such as naïve Bayes and logistic regression for supervised SFP. In terms of experimentation, the PROMISE and NASA MDP repositories are frequently used for conducting research. These findings are valuable in assisting the selection of appropriate modeling algorithms for our own experimentation. However, the articles to find out the accompanying metrics of Halstead metrics suite in SFP have been discussed below along with the summary in Table 1.

*Chiu (2011)* reported the classification accuracy of Halstead, when used with McCabe, LoC, and Branch count. The modeling has been performed using four different classification algorithms, *i.e.,* LR, SVM, ANN, and DIN The experiment on KC2 dataset shows the best results when used IDN for modeling.

*Dejaeger, Verbraken & Baesens (2013)* includes LR, RF, and the Bayesian Network (BN) classifiers for modeling on 11 public datasets. Halstead metrics suite along with McCabe

**Table 1  Summarized view of studies using Halstead metric suite for SFP.**

| Article | Metrics | Dataset | Technique | Performance measure |
|---|---|---|---|---|
| *Chiu (2011)* | Halstead, McCabe, LOC, Branch Count | KC2 | LR, SVM, ANN, Integrated decision network approach (IDN) | Acc, Pre, Recall, F-measure |
| *Dejaeger, Verbraken & Baesens (2013)* | Halstead, McCabe, LOC | JM1, KC1, MC1, PC1, PC2, PC3, PC4, PC5, EC12.0a, EC12.1a, EC13.0a | LR, RF, NB | AUC, H-Measure |
| *Arar & Ayan (2015)* | Halstead, McCabe | KC1, KC2, JM1, PC1, CM1 | ANN, Artificial Bee Colony (ABC) | AUC, Acc |
| *Dhanajayan & Pillai (2017)* | Halstead, McCabe, LOC, Branch Count | CM1 | NB, RF, ANN, Spiral life cycle model-based Bayesian classification (SLMBC) | False Negative Rate, False Positive Rate, Overall error rate |
| *Bhandari & Gupta (2018)* | Halstead, McCabe, LOC | JM1, PC1, KC1, jEdit | RF, DT, NB, SVM, ANN | Acc, F1-Score, precision, recall, AUC |
| *Shippey, Bowes & Hall (2019)* | Halstead, McCabe, LOC, Branch Count | T2, T1, EJDT, ArgoUML, AspectJ, JMOL, GenoViz, K Framework, SocialSDK, JMRI, JBoss Reddeer | NB, DT, RF. | Recall, Pre |
| *Ahmed et al. (2020)* | Halstead, McCabe, LOC, Branch_Count, Call_Pairs | PC1, PC2, PC3, PC4, PC5, JM1, KC1, MC1, Ecl2.0a, Ecl2.1a, Ecl3.0a, | DT, NB, SVM, RF, KNN, LR, | AUC |
| *Cetiner & Sahingoz (2020)* | Halstead, McCabe, LOC | PC1, JM1, KC1, CM1, KC2 | DT, NB, KNN, SVM, RF, MLP, Extra Trees, Ada boost, Gradient Boosting, Bagging | Acc |
| *Kumar, Kumar & Mohapatra (2021)* | Halstead, McCabe, LOC | PC1, PC2, PC3, PC4, CM1, JM1, KC3 DT, CRV, BN, LS, LR | LR, NB, D,T MLP, SVM, RF, LSSVM | Acc, AUC, F1-Score |

and LoC has been used as an Independent variable (IV) The results, both in terms of the AUC and H-measure have been recorded wherein NB outperforms.

*Arar & Ayan (2015)* utilized artificial neural networks (ANN) and the ABC optimization algorithm to analyze five datasets from the NASA Metrics Data Program repository. The classification approach was evaluated based on several performance indicators, including accuracy, probability of detection, probability of false alarm, balance, Area Under Curve (AUC), and Normalized Expected Cost of Misclassification (NECM). Halstead and McCabe metrics were employed as independent variables (IV). The experimental findings demonstrated the successful creation of a cost-sensitive neural network through the application of the ABC optimization algorithm.

*Dhanajayan & Pillai (2017)* assess the SFP capability of Halstead, McCabe, LOC, and Branch Count on CM1 data set using NB, RF, ANN, Spiral life cycle model-based Bayesian classification (SLMBC). The performance has been evaluated using false negative rate, false positive rate, and overall error rate.

*Bhandari & Gupta (2018)* proposes a spiral life cycle model-based Bayesian classification technique for efficient SFP and classification. In this process, initially, the independent software modules are identified which are Halstead, McCabe, and LoC. The experiment results show that RF achieves higher accuracy, precision, recall, probability of detection, F-measure, and lower error rate than the rest of the techniques.

*Shippey, Bowes & Hall (2019)* employed the utilization of Abstract Syntax Tree (AST) n-grams to detect characteristics of faulty Java code that enhance the accuracy of defect prediction. Various metrics such as Halstead, McCabe, Lines of Code (LoC), and Branch Count have been applied to train naïve Bayes, J489, and random forest models. The outcome reveals a strong and statistically significant correlation between AST n-grams and faults in certain systems, demonstrating a substantial impact.

*Ahmed et al. (2020)* proposed a software defect predictive development model using machine learning techniques that can enable the software to continue its projected task. Halstead, McCabe, LoC, Branch count and Call pairs have been used for modeling SVM, DT, NB, RF, KNN, and LR on three defect datasets in terms of f1 measure. The experiment results are in favor of LR.

*Cetiner & Sahingoz (2020)* conducted a comparative analysis of machine learning-based software defect prediction systems by evaluating 10 learning algorithms including decision tree, naïve Bayes, k-nearest neighbor, support vector machine, random forest, extra trees, ada boost, gradient boosting, bagging, and multi-layer perceptron. The analysis was performed on the public datasets CM1, KC1, KC2, JM1, and PC1 obtained from the PROMISE warehouse. Halstead, McCabe, and LoC were utilized for modeling the classification algorithms. The experimental findings demonstrated that the Random Forest (RF) model exhibited favorable accuracy levels in software defect prediction, thus enhancing the software quality.

*Kumar, Kumar & Mohapatra (2021)* aimed to create and compare different SFP models using Least Squares Support Vector Machine (LSSVM) with three types of kernels: Linear, Polynomial, and Radial Basis Function (RBF). These models aim to classify software modules as either faulty or non-faulty based on various software metrics such as Halstead software metrics, McCabe, and Lines of Code (LoC). To assess the impact of the proposed models, experiments are conducted on fifteen open source projects. The performance of the models is evaluated using Accuracy, F-measure, and ROC AUC as metrics. The experimental findings indicate that the LSSVM model with a polynomial kernel outperforms the LSSVM model with a linear kernel, and performs similarly to the RBF kernel. Additionally, the models developed using LSSVM demonstrate improved accuracy in SFP compared to commonly used models in the field.

The studies have explored the use of different classification algorithms and metrics along with Halstead for SFP. The studies utilized Halstead, McCabe, LoC, and Branch count metrics as independent variables IV for modeling. LR, MLP, RF, naïve Bayes (NB), Decision Tree classification (SLMBC), and LSSVM with Linear, Polynomial, and Radial Basis Function (RBF) kernels were among the most algorithms used. Performance evaluation metrics included AUC, accuracy, precision, recall, F-measure, overall error rate, and ROC AUC.

# DECOMPOSITION OF HALSTEAD BASE METRICS

In the ML domain, feature decomposition is an activity to break down a feature into smaller features. The decomposition enables ML algorithms to comprehend features and thus improves model performance by uncovering potential information (*Caiafa et al., 2020*). In this article, we decompose the Halstead base metrics by breaking them down into smaller metrics. The Halstead metric suite comprises four base metrics and seven derived metrics which are derived out of the base metrics. The four base metrics are Total operators (N1), Total operands (N2), Unique operators (n1), and Unique operands (n2). Likewise, these base metrics are composed of operators and operands. According to the definition of Halstead, the software program is a composition of tokens. Each token can either be an operator or an operand. Operand includes variables and constants. While all other tokens are included in operators that can safely be extracted from various languages (*Halstead, 1972*; *Govil, 2020*). The following subsection would elaborate on the decomposition of Halstead operators and Halstead operands.

## Decomposition of Halstead operators

According to the sources (*Python Academy, 2020*; *Pratt, 2021*; *Feroz, 2019*; *Gustedt, 2019*; *Gvero, 2013*), it can be stated that the definition of Halstead operator is more comprehensive than that of the operators defined in conventional programming languages such as C, C++, Java, and others. Halstead operators include not only the commonly recognized operators such as brackets, semicolon, colon, and punctuation marks but also function names and other elements that are not considered operators in conventional languages. It should also be noted that there are some operators present in some programming languages that are not present in others. For instance, the increment/decrement operators are present in C, C++, Java, and PhP but are not present in Python. Therefore, it can be concluded that while some Halstead operators are recognized as operators by all major programming languages, others are not defined as operators by all languages or only by a few languages. Consequently, five overarching categories can be discerned that are universally applicable to all major programming languages.

1. **Assignment operators**, are operators that are explicitly declared as assignment operators in conventional programming languages. The assignment operation can be performed explicitly using the $=$ operator, or in combination with other operators such as $+=$ and $-=$, or sometimes implicitly using operators like $--$ and $++$.
2. **Arithmetic operators**, are a class of operators that are conventionally recognized as such in programming languages. These operators, such as addition, subtraction, and multiplication, can be explicitly declared using symbols like $+$ and $-$, and are commonly used in mathematical expressions.
3. **Logical operators** are formally defined and recognized in conventional programming languages. The conjunction operator, denoted by && (AND), and the disjunction operator, denoted by || (OR), are examples of such operators.
4. **Relational operators** are defined as binary operators in traditional programming languages and satisfy the properties of reflexivity, anti-symmetry, and transitivity, such

**Table 2  Halstead operators with their corresponding decomposed operators.**

| Base metrics | Decomposed metrics | Description |
|---|---|---|
| | As | Total assignment operators |
| Total operators (N1) | A | Total arithmetic operators |
| | R | Total relational operators |
| | Log | Total logical operators |
| | O | Total operators other than assignment, arithmetic, relational, and logical operators |
| | as | Unique assignment operators |
| Unique operators (n1) | a | Unique arithmetic operators |
| | r | Unique relational operators |
| | log | Unique logical operators |
| | o | Unique operators other than assignment, arithmetic, relational, and logical operators |

as the less than or equal to operator ($<=$) and the greater than or equal to operator ($>=$).

5. **Others** The remaining Halstead operators include items such as brackets and function names, among others.

Table 2 presents the Halstead operators along with their corresponding decomposed metrics. For example, if $A$ represents the total operator, it can be decomposed into $a_1, a_2, \ldots, a_n$ such that the value of $A$ equals the sum of the values of the decomposed operators $a_1, a_2, \ldots, a_n$. In this article, we present a demonstration of the Decomposed Halstead base metric utilizing the Java programming language. Within Java, the operators that are declared can be categorized into the five previously discussed categories. An overview of the Java operators, along with their corresponding proposed categories, is provided in Table 3. Furthermore, we reiterate that the objective of this article is to showcase the application and effectiveness of the Decomposed Halstead base metric using Java.

## Decomposition of Halstead operands

In the realm of traditional computer programming, the term "operand" is utilized to describe any object that possesses the capability to be manipulated. Halstead's operands can be decomposed into two distinct and mutually exclusive types: variables and constants. A variable refers to a data item whose value can be modified during the execution of a program. Conversely, a constant is a literal utilized to represent a fixed value within the source code. To illustrate, consider the following code snippet written in the Java programming language:

```
1. int x = 5;
2. final int y=10;
3. char z='b';
```

x, y, and z are variables, while 5, 10 and 'b' are constants, rest of the tokens *i.e.*, `int`, `final`, `char`, `=`, and `;` (semicolon) are all operators. Keeping in view such decomposition, if $B$ represents the total operands, it can be decomposed into $b_1, b_2, \ldots, b_n$ such that the

**Table 3  Java operators and their category.**

| Category | Operator | Description |
|---|---|---|
| Assignment | = | Simple assignment operator |
| | += | Addition and assignment operator |
| | −= | Subtraction and assignment operator |
| | *= | Multiplication and assignment operator |
| | /= | Divide and assignment operator |
| | %= | Modulus and assignment operator |
| | &= | Bitwise and assignment operator |
| | ^= | Bitwise exclusive OR and assignment operator |
| | \|= | Bitwise inclusive OR and assignment operator |
| | <<= | Left shift and assignment operator |
| | >>= | Right shift and assignment operator |
| | ++ | Increment operator |
| | −− | Decrement operator |
| Arithmetic | + | Additive operator |
| | − | Subtraction operator |
| | * | Multiplication operator |
| | / | Division operator |
| | % | Remainder operator |
| | + | Unary plus operator |
| | − | Unary minus operator |
| Logical | && | Conditional-AND |
| | \|\| | Conditional-OR |
| | ! | Logical complement operator |
| Relational | == | Equal to |
| | > | Greater than |
| | != | Not equal to |
| | >= | Greater than or equal to |
| | < | Less than |
| | <= | Less than or equal to |
| Others | Function name | Any function name in a program |
| | Class name | Any class name in a program |
| | {,}, [,] | Brackets |
| | int, float, double, *etc.* | Data types |
| | ; , | Special symbols |

value of $B$ equals the sum of the values of the decomposed operands $b_1, b_2, \ldots, b_n$. The Halstead operands along with corresponding decomposed metrics are shown in Table 4.

The decomposition of operators and operands is a universally applicable principle that encompasses all prominent programming languages, such as Java, C, C++, and others. This decomposition entails the systematic categorization of operators and operands into distinct and non-overlapping classes, as depicted in Fig. 1.

We employ Java projects as a means of experimentation in this study. In order to enhance comprehension regarding the decomposition of operators and operands, we

**Table 4  Halstead operands with their corresponding decomposed operands.**

| Base metrics | Decomposed metrics | Description |
|---|---|---|
| Total operands (N2) | Var | Total variables |
| | C | Total constants |
| Unique operands (n2) | var | Unique constants |
| | c | Unique variables |

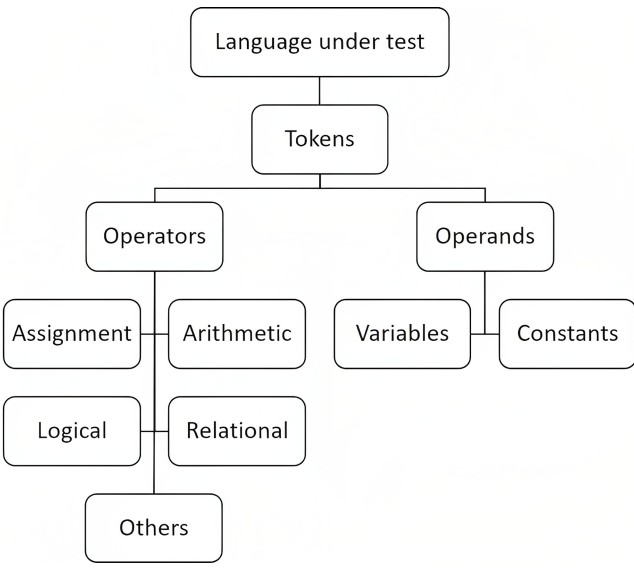

**Figure 1  Hierarchy of decomposed operators and operands.**

present a concise illustrative Java code snippet, as depicted in Fig. 2. Table 5 provides a comprehensive display of the assigned values for each token, corresponding to their respective decomposed operator or operand class.

## METHODOLOGY

Our objective is to evaluate the impact of the decomposed Halstead base metric in SFP. To achieve this goal, we have designed a methodology as depicted in Fig. 3, which would steer the execution of our experiment, elaborated in the next section. In the proposed methodology, our initial step involves the selection of case studies. The optimal choice for a case study would encompass a publicly available dataset along with accompanying source code. These case studies will serve as the foundation for the development of three distinct datasets. The first dataset, denoted as "Dataset-1", will encompass the Halstead metric suite as well as frequently reported valuable metrics utilized in SFP, such as Lines of Code (LoC) and McCabe. The second dataset, referred to as "Dataset-2", will consist of the same software metrics employed in "Dataset-1", with the exception of the Halstead base metrics. To obtain the Decomposed Halstead base metrics, the source code of the selected case studies will be parsed using a metrics extractor. The parsed Decomposed Halstead base

```
1   void merge(int a[], int beg, int mid, int end)
2   {
3       int i, j, k, n1 = mid - beg + 1, n2 = end - mid;
4       int LeftArray[n1], RightArray[n2];
5       for (int i = 0; i < n1; i++)
6       LeftArray[i] = a[beg + i];
7       for (int j = 0; j < n2; j++)
8       RightArray[j] = a[mid + 1 + j];
9       i = 0; j = 0; k = beg;
10      while (i < n1 && j < n2){
11          if(LeftArray[i] <= RightArray[j]){
12              a[k] = LeftArray[i];
13              i++;
14          }
15          else{
16              a[k] = RightArray[j];
17              j++;
18          }
19          k++;
20      }
21      while (i<n1){
22          a[k] = LeftArray[i];
23          i++; k++;
24      }
        while (j<n2){
            a[k] = RightArray[j];
            j++; k++;
        }
    }
```

**Figure 2** Sample code for Decomposed Halstead base metrics demonstration.

**Table 5** Metrics count for sample code.

| Halstead | Decomposed Halstead | Tokens |
|---|---|---|
| | As: 22 | =: 13 ++:9 |
| | A: 6 | -: 2 +: 4 |
| N1: 142 | R: 7 | <: 6 <=: 1 |
| | Log: 1 | &&: 1 |
| | O: 106 | void: 1 mrge: 1 (: 7 ): 7 int: 7 {: 6 {: 6 ,: 8 ;: 22 [: 17 ]: 17 for: 2 while: 3 if:1 else: 1 |
| | as: 2 | =: 13 ++:9 |
| | a: 2 | -: 2 +: 4 |
| n1: 22 | r: 2 | <: 6 <=: 1 |
| | log: 1 | &&: 1 |
| | o: 15 | void: 1 mrge: 1 (: 7 ): 7 int: 7 {: 6 {: 6 ,: 8 ;: 22 [: 17 ]: 17 for: 2 while: 3 if:1 else: 1 |
| | Var: 64 | beg: 4 mid: 4 end: 3 I: 12 j: 12 k: 9 n1: 5 n2: 5 LeftArray: 5 RightArray: 5 |
| N2: 66 | C: 2 | 1: 1 0: 1 |
| | var: 10 | beg: 4 mid: 4 end: 3 I: 12 j: 12 k: 9 n1: 5 n2: 5 LeftArray: 5 RightArray: 5 |
| n2: 12 | c: 2 | 1: 1 0: 1 |

metric will then be merged with "Dataset-2", resulting in the creation of a new dataset called "Dataset-3". This new dataset, "Dataset-3", will encompass both the decomposed Halstead base metric and the SFP metrics selected in "Dataset-2". Subsequently, a machine learning algorithm will be employed to model the relationship between the independent variable

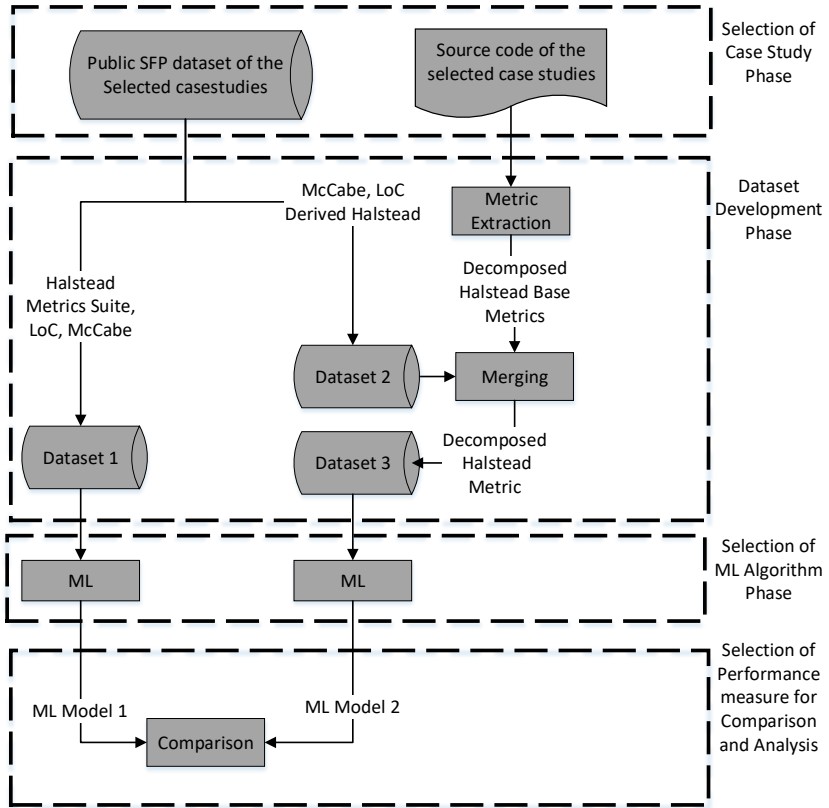

**Figure 3  Methodology.**

(IV) and the dependent variable (DV) in both "Dataset-1" and "Dataset-3". Finally, the performance of "ML Model-1" and "ML Model-2" will be compared and analyzed utilizing various performance measures. In summary, the methodology comprises the following key phases:

1. Selection of case studies
2. Datasets' development
3. Selection of ML algorithm
4. Selection of performance measures for comparison and analysis

## Selection of case studies

In this phase, a selection of case studies would be made, upon which subsequent processing shall be performed. It is widely acknowledged that ML-based empirical studies exhibit a high degree of bias due to the quality of data. This is largely attributed to the inadequacy of data and the absence of systematic data collection procedures. It is noteworthy that SFP has been executed using a diverse range of datasets, which may be classified into four categories based on their availability, namely: private, partially private, partially public, and public, as per *Radjenović et al. (2013)*. In private datasets, neither the source code nor the fault information is provided, rendering studies based on these datasets non-repeatable. Partially private datasets offer access only to source code and/or metrics values, without

**Table 6  Types of datasets w.r.t. availability of metrics values, fault information, and source code.**

| Type of dataset | Metrics' values | Fault information | Source code |
|---|---|---|---|
| Private | ✗ | ✗ | ✗ |
| Partially private | ✓ | ✓ | ✗ |
| Partially public | ✗ | ✓ | ✓ |
| Public | ✓ | ✓ | ✓ |

fault information. Partially public datasets typically provide access to both source code and fault data, but not metrics values, which must be extracted from the source code and mapped to fault data from the repository (*Radjenović et al., 2013*). Public datasets, on the other hand, refer to datasets in which metrics values, source code, and fault data are publicly available for all modules in a software system. Table 6 illustrates these five dataset types.

Since we aim to evaluate the performance disparity between Halstead base metrics and Decomposed Halstead base metrics, a suitable dataset for this task would be one that is at least partially publicly accessible.

## Datasets' development

After the identification of the case studies during the phase of case study selection, the subsequent task is dataset development. As previously discussed, the appropriate case studies for our experiment should possess publicly available software metrics datasets and their corresponding source code. In order to fulfill this requirement, we proceed to construct Dataset-1 using the chosen public dataset, specifically selecting the Halstead metric suite along with McCabe and LoC metrics. For the development of Dataset-2, we select all SFP metrics from the public dataset, which were previously included in dataset-1, excluding the Halstead base metrics. There is no publicly available dataset containing information on decomposed Halstead base metrics. Therefore, we will employ a metrics extractor on the source code of selected case studies to obtain these metrics. Existing metrics extractors have been examined for the purpose of calculating the decomposed Halstead base metrics. However, it has been noted that these existing extractors possess three primary limitations:

1. The extractors have a lack of extensibility and, hence, may not be used to integrate with existing frameworks/extractors.
2. The extractors are metrics-specific and may not extract new metrics. Hence they are not easy to adapt to other metrics.
3. The extractors have an ambiguous interpretation of some metrics. Hence, more than one variant of the same metric exists which is reported in *Nilsson (2019)*.

Taking into consideration the constraints and requirements of our experimentation, it is imperative to undertake the development of a custom-built extractor that possesses the capability to extract Halstead base metrics from the given source code. The primary function of our extractor involves parsing the source code of the designated case studies with the objective of extracting decomposed Halstead base metrics. Throughout the process of parsing the Decomposed Halstead metrics suite, a hierarchical tree, as illustrated

in Fig. 1, will be employed. Where program statements will first be split into tokens, and then classified into operands and operators. The process will be sequential, with operands containing variables and constants being considered first, followed by operators. Within the operators, assignment, arithmetic, logical, and relational operators will be identified, while the remaining operators will be categorized as "others".

Once the decomposed Halstead base metrics have been extracted, they are to be merged with Dataset-2 in a formal manner. During the parsing phase, it is imperative to preserve the information pertaining to the "Complete path of the source code file" as well as the "Name of Class" contained within that file. This information plays a crucial role in distinguishing between similar class names across multiple files and different classes within the same file. Dataset-2 also encompasses this essential information. The merging process culminates in the creation of Dataset-3, which encompasses the parsed Decomposed Halstead base metrics and the specifically chosen SFP metrics from Dataset-2. Subsequently, Dataset-1 and Dataset-2 will serve as inputs to the machine learning algorithm for the purposes of model construction and the execution of SFP.

## ML algorithm

Once the datasets have been developed, it is important to determine the suitable ML algorithm for modeling. ML algorithms offer a range of options for various tasks. Even in SFP literature, various ML algorithms have been employed with varying performance. Linear regression and logistic regression are used for regression and binary classification, respectively. Decision trees create a tree-like model for decision-making, while random forests combine multiple decision trees. SVM finds hyperplanes for class separation, and naïve Bayes applies probabilistic reasoning. K-Nearest Neighbors classifies instances based on similarity, and neural networks learn complex patterns. Gradient boosting combines weak models, while clustering algorithms group similar instances. Dimensionality reduction techniques reduce features, and reinforcement learning involves agents maximizing rewards. These algorithms represent a subset of ML methods, and selecting the right one depends on the specific task and data constraints.

## Selection of performance measures for comparison and analysis

The results of ML models are assessed by some performance measures. In classification, various performance measures are employed based on the task and data. Accuracy calculates the ratio of correct predictions, while precision focuses on the accuracy of positive predictions. Recall measures the proportion of true positive predictions out of all actual positives. The F1 score balances precision and recall. Specificity measures true negatives, important in reducing false alarms. AUC-ROC evaluates binary classifiers' ability to rank instances correctly.

## EXPERIMENTATION

Considering the methodology, our experiment encompasses four fundamental components, namely, case studies, sets of variables (datasets), modeling algorithms, and analysis of ML models using performance measures.

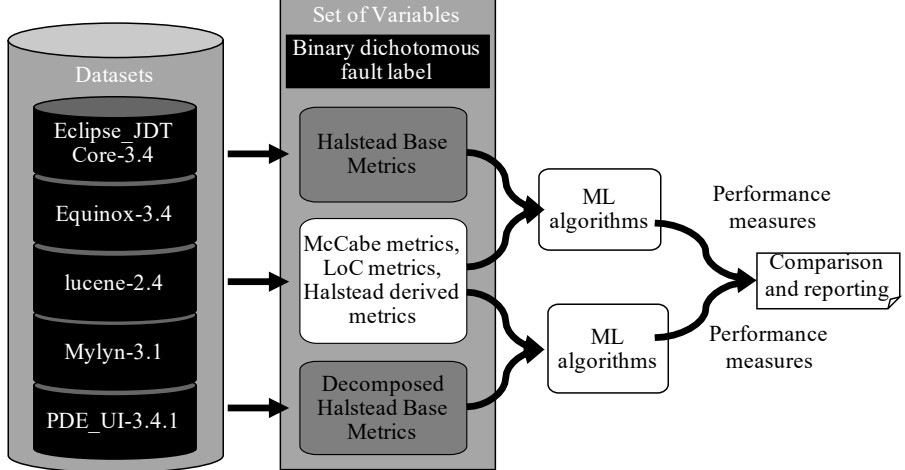

**Figure 4** **Experimental design.**

1. Regrettably, the public datasets do not encompass any information pertaining to the Decomposed Halstead base metrics. As a result, we undertake the task of constructing datasets by meticulously extracting the requisite metrics and bug-related data from the source code. Consequently, we carefully select five software projects, as expounded upon in the 'Case Study' section, which possess both the source code and the necessary fault information.

2. The evaluation of the Decomposed Halstead base metrics has been carried out through the implementation of two experiments. Experiment 1 encompasses the Halstead metric suite, coupled with the McCabe and Loc metrics. On the other hand, experiment 2 comprises the Decomposed Halstead base metrics, along with the Halstead derived, McCabe, and LoC metrics.

3. According to the literature review, the commonly employed ML algorithms are identified, while the Halstead metric suite is utilized within the SFP context. These aspects are further expounded upon in 'ML modeling' section.

4. The evaluation of the results has been carried out by using Accuracy, F-measure, and AUC.

   The graphical representation of our experiments is shown in Fig. 4.

## Case study

For experimentation purposes, we selected the following five datasets for their public source code along with fault information.

1. Apache Lucene 2.4 (lucene.apache.org)
2. Eclipse equinox framework 3.4 (www.eclipse.org/equinox/)
3. Eclipse JDT Core 3.4 (www.eclipse.org/jdt/core/)
4. Eclipse PDE UI 3.4.1 (www.eclipse.org/pde/pde-ui/)
5. Mylyn 3.1 (www.eclipse.org/mylyn/)

These object-oriented based projects are developed in Java and are publicly available. *Tóth, Gyimesi & Ferenc (2016)* assigned fault labels using a bug tracking system.

**Apache Lucene** is a Java-based text search engine library that offers exceptional performance and comprehensive features. It is well-suited for a wide range of applications that necessitate full-text search capabilities, particularly those that span multiple platforms. Apache Lucene is an open-source project that can be freely downloaded.

**Eclipse JDT Core** is a component of the Eclipse Java Development Tools (JDT) project. JDT is a collection of plug-ins for the Eclipse Integrated Development Environment (IDE) that provides a comprehensive set of features for Java development. JDT Core specifically focuses on the core functionality of Java development within Eclipse. It provides the infrastructure and APIs necessary for working with Java source code, such as parsing, analyzing, and manipulating Java code. It forms the foundation for many Java-related features in Eclipse, including code editing, refactoring, debugging, and code generation.

**Eclipse PDE UI** provides a comprehensive set of tools to create, develop, test, debug and deploy Eclipse plug-ins. PDE UI also provides multi-page editors that centrally manage all manifest files of a plug-in or feature. It carries new project creation wizards to create a new plug-in, fragment, feature, feature patch, and update sites.

**Eclipse equinox** is an implementation of the OSGi core framework specification, a set of bundles that implement various optional OSGi services and other infrastructure for running OSGi-based systems. It is responsible for developing and delivering the OSGi framework implementation used for all of Eclipse. The Equinox OSGi core framework implementation is used as the reference implementation. The goal of the Equinox project is to be a first-class OSGi community and foster the vision of Eclipse as a landscape of bundles.

**Mylyn** is the task and application life cycle management framework for Eclipse. It provides a revolutionary task-focused interface and a task management tool for developers.

## Set of variables

In our experimentation, the *DV* is a binary dichotomous fault label, *i.e., fp* and *nfp*. Since the selected dataset contains the numerical fault label, we transformed it into binary using the following rulings shown in Eq. (1).

$$Label = \begin{cases} fp & No.\,of\,faults > 0 \\ nfp & otherwise \end{cases} \tag{1}$$

The distribution of fault/fault-free is shown in Fig. 5.

The **IV** comprise five metrics sets, *i.e.,* Halstead base metrics, Halstead derived metrics, LoC metric suite McCabe metric suites, and Decomposed Halstead base metrics. These metrics set are placed in three distinct sets as follows:

Set 1: {Halstead base metrics}
Set 2: {McCabe, LoC , Halstead derived metrics}
Set 3: {Decomposed Halstead Base Metrics}

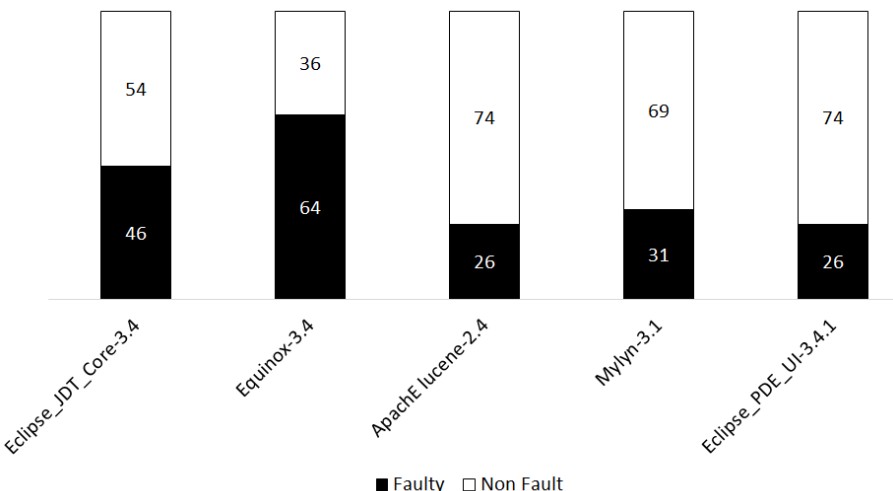

**Figure 5** Distribution of fault and fault free in the five datasets.

**Table 7** Metrics in set-1.

| Metrics set | Metric name | Short form | Description |
|---|---|---|---|
| Halstead- | OPERATORS | N1 | Total operators |
| Base- | OPERANDS | N2 | Total operands |
| Metric | UNIQUE OPERATORS | n1 | Unique operators |
| | UNIQUE OPERANDS | n2 | Unique operands |

The first experiment comprises the elements of Set-1 and Set-2, while the second experiment comprises the elements of Set-2 and Set-3. The detailed description and their distribution in the experiments are shown in Tables 7, 8 and 9.

## Data preprocessing

The data preprocessing comprises the following two steps

1. Conversion of numerical fault label to binary fault label as shown in Eq. (1).
2. Assessing the validation of data. In this activity, we analyse and ensure the absence of missing values, out-of-range values, Null values, invalid values (like negative in total operators), *etc.*

## ML modeling

The literature review section highlights the frequent utilization and effectiveness of six ML algorithms, establishing a foundation for incorporating these algorithms into our experiments as well.

1. *Logistic regression* model is a statistical approach that estimates the likelihood of one event occurring, given two possible outcomes. It achieves this by expressing the log-odds (logarithm of the odds) of the event as a linear combination of one or more independent variables (IV). Specifically, in binary logistic regression, there is a

**Table 8  Metrics in set-2.**

| Metrics set | Metric name | Short form | Description |
|---|---|---|---|
| | Assignment operators | As | Total assignment operators |
| | Unique assignment operators | as | Unique assignment operators |
| | arithmetic operators | A | Total arithmetic operators |
| | Unique arithmetic operators | a | Unique arithmetic operators |
| Decomposed | Relational operators | R | Total relational operators |
| Halstead- | Unique relational operators | r | Unique relational operators |
| Base- | Logical operators | Log | Total logical operators |
| Metrics | Unique logical operators | log | Unique logical operators |
| | Other operators | O | Total operators other than arithmetic, logical, and relational operators |
| | Unique other operators | o | Unique operators other than arithmetic, logical, and relational operators |
| | Variables | Ver | Total variable |
| | Unique variables | ver | Unique variable |
| | Constants | C | Total constant |
| | Unique constants | c | Unique constant |

**Table 9  Metrics in set-3.**

| Metrics set | Metric name | Short form | Description/Formula |
|---|---|---|---|
| Halstead | Halstead length | N | $N = N1 + N2$ |
| derived | Halstead level | L | $L = V^*/V$ |
| | Halstead difficulty | D | $D = 1/L$ |
| | Halstead volume | V | $V = N * \log2(n1+n2)$ |
| | Halstead effort | E | $E = V/L$ |
| | Halstead prog time | T | $T = E/18$ |
| | Halstead error est | B | $B = E2/3/1000$ |
| LoC | Loc total | LoC | The total number of lines for a given module |
| Metrics | Loc executable | LoEx | The number of lines of executable code for a module (not blank or comment) |
| | Loc blank | LoB | The number of blank lines in a module |
| | Loc comments | LoCm | The number of lines of comments in a module |
| | Loc code & comment | LoCoCm | The number of lines which contain both code and comment in a module |
| McCabe | Cyclomatic complexity | v(G) | $v(G) = e - n + 2$ |
| Metrics | Design complexity | iv(G) | The design complexity of a module |
| | Essential complexity | ev(G) | The essential complexity of a module |

solitary dependent variable that takes on binary values, labeled as "0" and "1", and is represented by an indicator variable. Meanwhile, the IV can be a continuous variable.

2. *Multilayer perceptron (MLP)* is a fully connected class of feed-forward artificial neural network (ANN). An MLP consists of at least three layers of nodes: an input layer, a hidden layer, and an output layer. Except for the input nodes, each node is a neuron that uses a nonlinear activation function. MLP utilizes a supervised learning technique called backpropagation for training. Its multiple layers and non-linear activation distinguish MLP from a linear perceptron. It can distinguish data that is not linearly separable. MLP is divided into an input layer, an output layer and a hidden layer. The information is collected through the input layer, and the data is input into the hidden layer for analysis and processing. This study uses an MLP model with a single hidden layer, and the initial learning rate is 0.3. *Pinkus (1999)*.

3. *Naïve Bayes* classifiers employ a probabilistic approach to classification. They exhibit excellent scalability, with the number of parameters directly proportional to the number of variables (features/predictors) in the learning problem. Training using maximum-likelihood estimation can be efficiently achieved through a closed-form expression, resulting in linear time complexity, unlike the computationally intensive iterative approximation methods employed by various other classifiers. *Garg (2013)*.

4. *Decision trees* categorized instances through the arrangement of their feature values in a process known as classification. This involves using a decision tree where each node represents a feature in the instance being classified, and each branch corresponds to a possible value of that node. Starting from the root node, the instances are classified and sorted based on their feature values. Decision tree learning, which is a technique utilized in data mining and machine learning, makes use of a decision tree as a predictive model. It connects information about an item to draw conclusions about the item's target value. Post-pruning techniques are commonly employed to improve the performance of decision tree classifiers. These techniques involve evaluating the decision tree's performance using a validation set and removing any node that does not contribute significantly. The removed node is then assigned the most frequent class among the training instances associated with it. In this study, a model based on decision trees is employed, using the C5.0 algorithm with a minimum number of leaf nodes. This choice addresses the problem of excessive branches that can occur with the ID3 algorithm. Furthermore, during the construction of the decision tree, pruning is performed to discretize continuous data, with the limit set to the maximum number of leaf nodes.

5. *Random forests* an ensemble learning technique used for classification, consists of multiple decision trees whose outputs are combined through majority voting. For regression tasks, the ensemble returns the mean or average prediction from the individual trees. Random forests typically surpass decision trees in terms of performance. *Biau & Scornet (2016)*.

6. *Support vector machines (SVMs)* are closely connected to classical MLP neural networks. SVMs are centered around the concept of a margin, which exists on either side of a hyperplane that separates two classes of data. By maximizing this margin, the goal is to create the largest possible distance between the separating hyperplane and

the instances on both sides, which has been proven to reduce the upper bound on the expected generalization error (*Kotsiantis, 2007*). The SVM-based model utilizes the Gaussian inner product as the kernel function, known as SVM-Kernel. By iteratively solving sub-problems, the prediction of large-scale problems is ultimately achieved. In this particular model, the gamma parameter is set to 0.024.

$$y(x) = w^T \Phi(x) + c \tag{2}$$

where $x$ is the input vector and $y$ is the output vector. $\phi(x)$ is a polynomial kernel function. $w$ and $c$ represent the adjusted weight vector and scalar threshold values, respectively.

## Evaluation measure

For evaluation purposes, we take three performance measures for the assessment of models. These performance measures are Accuracy, AUC, and F-measure (*Rizwan, Nadeem & Sindhu, 2019*).

1. Accuracy shows the correct predictions. It is a good measure when the classes in the test dataset are nearly balanced. It measures the ability of a classifier in correctly identifying all samples, no matter if it is positive or negative.

$$Accuracy = \frac{TP + TN}{P + N} \tag{3}$$

2. The F-measure is a harmonic mean of precision and recall. F-measure can mathematically be written as follows

$$Fmeasure = \frac{2 \times Precision \times Recall}{Precision + Recall} \tag{4}$$

3. The AUC stands for Area under the Receiver Operating Characteristics Curve, which is a curve that represents the probability and measures the level of separability. It indicates the model's ability to differentiate between classes. AUC values range from 0 to 1. A value of 0.0 for AUC indicates 100% incorrect prediction, while a value of 1.0 represents 100% accurate prediction.

The performance of the prediction models is validated using 10-fold cross-validation. During cross-validation, the input dataset is divided into 10 equal-sized folds in a random manner. Out of these, nine folds are utilized for training the model, while the remaining fold is used for testing the model. This process is repeated 10 times, with each iteration excluding a different fold for testing purposes.

## RESULTS AND DISCUSSION

In this section, we shall present the findings of our two distinct experiments, namely Experiment-1 and Experiment-2. The results of both experiments have been evaluated based on three performance metrics, namely Accuracy, F-measure, and AUC. Notably, the outcomes obtained in Experiment-2 are observed to be superior to those reported in Experiment-1. This clearly indicates a significant improvement in the performance of fault prediction using Decomposed Halstead base metrics. In terms of algorithms SVM performs well in three datasets, *i.e.,* Apache Lucene, Eclipse JDT Core, and Equinox framework. The

obvious reason is the clear margin of separation between classes in these datasets. Along with high dimensional spaces and the least difference in the number of dimensions and the number of samples. NB performs well in the rest of two datasets, *i.e.,* Eclipse PDE UI and Mylyn. This is due to the reason that there exists the least correlation between features. Moreover, there are fewer training examples, *i.e.,* less than 1,000 samples in Eclipse PDE and less than 1,400 samples in Mylyn. Likewise, there exists well-formed discretization of our numerical data in these two datasets, which makes NB perform better than the rest of the algorithms. The results have been shown in three different ways, *i.e.,* weightage of individual metric with information gain (Table 10), detail results (Tables 11, 12 and 13), and difference in the results (Figs. 6, 7 and 8).

The data presented in Table 10 reveals distinct patterns in terms of information gain among various metrics. Notably, both the McCabe and LoC metrics stand out by demonstrating notably better information gain compared to the others. Concurrently, the Halstead decomposed metrics also showcase a relatively high level of information gain, capturing attention. In contrast, the metrics derived from the Halstead approach exhibit the lowest information gain among all the metrics under consideration. This observation aligns closely with the consensus within the broader research community. It underscores the idea that metrics such as McCabe, LoC, and the decomposed Halstead metrics play a significant role in providing valuable information and meaningful insights. Meanwhile, metrics stemming from the Halstead-derived methods appear to offer comparatively limited utility when it comes to facilitating substantial information gain.

In Tables 11, 12 and 13 the first column identifies the datasets, and the subsequent columns present the algorithms used for model building in Experiment 1 and 2. Each row contains the result of the datasets in the corresponding algorithm. The maximum Accuracy achieved, according to Table 11, 0.97 in Apache Lucene 2.4, 0.89 in Eclipse equinox framework 3.4, 0.97 in Eclipse JDT Core 3.4 using SVM, 0.98 in Eclips PDE UI 3.4.1, and 0.95 in Mylyn 3.1 has been observed using NB. The maximum F-measure achieved, according to Table 12, 0.96 in Apache Lucene 2.4, 0.98 in Eclipse equinox framework 3.4, 0.99 in Eclipse JDT Core 3.4 using SVM, 0.98 in Eclips PDE UI 3.4.1 and 0.99 in Mylyn 3.1 using NB have been observed. The maximum AUC achieved, according to Table 13, 0.98 in Apache Lucene 2.4, 0.97 in Eclipse equinox framework 3.4, 0.95 in Eclipse JDT Core 3.4 using SVM, 0.96 in Eclipse PDE UI 3.4.1 and 0.99 in Mylyn 3.1 using NB.

The difference in the performance observed in the two experiments is shown in figures, *i.e.,* Figs. 6, 7 and 8. The bars in the positive axis indicate the improvement achieved in Experiment-2.

In terms of Accuracy, the minimum difference is 0.025, which has been observed in the Eclipse equinox dataset using MLP. Whereas the maximum difference has been observed in the Eclipse PDE dataset, *i.e.,* 0.31 using DT.

**Table 10  Information gain of the metrics in the datasets.**

|  | Metrics | Apache lucene | Eclipse equinox framework | Eclipse JDT core | Eclipse PDE UI | Mylyn |
|---|---|---|---|---|---|---|
|  | v(g) | 0.72 | 0.77 | 0.62 | 0.71 | 0.63 |
| McCabe | iv(g) | 0.74 | 0.67 | 0.77 | 0.64 | 0.61 |
|  | ev(g) | 0.70 | 0.53 | 0.55 | 0.68 | 0.50 |
|  | LoC | 0.58 | 0.41 | 0.60 | 0.40 | 0.51 |
|  | LoEx | 0.63 | 0.53 | 0.48 | 0.54 | 0.60 |
| LoC | LoB | 0.46 | 0.38 | 0.24 | 0.47 | 0.32 |
|  | LoCm | 0.28 | 0.41 | 0.25 | 0.44 | 0.22 |
|  | LoCoCm | 0.55 | 0.61 | 0.53 | 0.41 | 0.53 |
| Halstead | N | 0.72 | 0.55 | 0.62 | 0.57 | 0.62 |
| Derived | L | 0.55 | 0.62 | 0.47 | 0.56 | 0.42 |
|  | D | 0.55 | 0.69 | 0.42 | 0.52 | 0.63 |
|  | V | 0.65 | 0.55 | 0.43 | 0.69 | 0.42 |
|  | E | 0.47 | 0.53 | 0.62 | 0.52 | 0.42 |
|  | T | 0.42 | 0.53 | 0.48 | 0.31 | 0.43 |
|  | B | 0.52 | 0.52 | 0.46 | 0.55 | 0.32 |
| Halstead | N1 | 0.66 | 0.53 | 0.61 | 0.64 | 0.62 |
| Base | n1 | 0.51 | 0.67 | 0.46 | 0.55 | 0.62 |
|  | N2 | 0.66 | 0.59 | 0.66 | 0.52 | 0.42 |
|  | n2 | 0.47 | 0.52 | 0.42 | 0.68 | 0.43 |
| Decomposed | As | 0.72 | 0.62 | 0.74 | 0.69 | 0.62 |
| Halstead | as | 0.56 | 0.68 | 0.51 | 0.62 | 0.53 |
|  | A | 0.53 | 0.57 | 0.69 | 0.57 | 0.53 |
|  | a | 0.59 | 0.43 | 0.63 | 0.59 | 0.42 |
|  | R | 0.79 | 0.71 | 0.71 | 0.74 | 0.72 |
|  | r | 0.62 | 0.57 | 0.46 | 0.53 | 0.62 |
|  | Log | 0.63 | 0.64 | 0.72 | 0.68 | 0.63 |
|  | log | 0.53 | 0.67 | 0.59 | 0.57 | 0.52 |
|  | O | 0.77 | 0.62 | 0.79 | 0.58 | 0.62 |
|  | o | 0.79 | 0.54 | 0.61 | 0.51 | 0.62 |
|  | Ver | 0.63 | 0.68 | 0.65 | 0.67 | 0.62 |
|  | ver | 0.54 | 0.62 | 0.51 | 0.58 | 0.53 |
|  | C | 0.56 | 0.52 | 0.57 | 0.59 | 0.52 |
|  | c | 0.47 | 0.44 | 0.43 | 0.44 | 0.41 |

In terms of F-measure, the minimum difference is 0.07, which has been observed in the Apache Lucene dataset using LR. Whereas the maximum difference has been observed in the Eclipse PDE dataset, *i.e.*, 0.26 using RF.

In terms of AUC the minimum difference is 0.07, which has been observed in Apache Lucene dataset using RF. Whereas the maximum difference has been observed in Eclipse Equinox dataset, *i.e.*, 0.25 using DT.

**Table 11  Accuracy in experiment 1 and 2.**

| Dataset | LR | | NB | | DT | | MLP | | RF | | SVM | |
|---|---|---|---|---|---|---|---|---|---|---|---|---|
| | Exp1 | Exp2 | Exp1 | Exp2 | Exp1 | Exp2 | Exp1 | Exp2 | Exp1 | Exp2 | Exp1 | Exp2 |
| Apache Lucene 2.4 | 0.75 | 0.83 | 0.66 | 0.88 | 0.75 | 0.91 | 0.7 | 0.91 | 0.76 | 0.94 | 0.82 | 0.97 |
| Eclipse equinox framework 3.4 | 0.71 | 0.83 | 0.72 | 0.82 | 0.69 | 0.79 | 0.65 | 0.91 | 0.76 | 0.83 | 0.79 | 0.89 |
| Eclipse JDT Core 3.4 | 0.73 | 0.93 | 0.68 | 0.89 | 0.66 | 0.91 | 0.79 | 0.82 | 0.67 | 0.94 | 0.81 | 0.97 |
| Eclipse PDE UI 3.4.1 | 0.72 | 0.88 | 0.79 | 0.98 | 0.65 | 0.86 | 0.69 | 0.92 | 0.65 | 0.9 | 0.76 | 0.87 |
| Mylyn 3.1 | 0.66 | 0.83 | 0.8 | 0.95 | 0.72 | 0.8 | 0.66 | 0.82 | 0.73 | 0.84 | 0.76 | 0.87 |

**Table 12  F-measure in experiment 1 and 2.**

| Dataset | LR | | NB | | DT | | MLP | | RF | | SVM | |
|---|---|---|---|---|---|---|---|---|---|---|---|---|
| | Exp1 | Exp2 | Exp1 | Exp2 | Exp1 | Exp2 | Exp1 | Exp2 | Exp1 | Exp2 | Exp1 | Exp2 |
| Apache Lucene 2.4 | 0.78 | 0.86 | 0.76 | 0.8 | 0.66 | 0.87 | 0.74 | 0.91 | 0.69 | 0.86 | 0.8 | 0.96 |
| Eclipse equinox framework 3.4 | 0.7 | 0.91 | 0.72 | 0.95 | 0.75 | 0.92 | 0.75 | 0.92 | 0.75 | 0.96 | 0.79 | 0.98 |
| Eclipse JDT Core 3.4 | 0.72 | 0.96 | 0.79 | 0.95 | 0.68 | 0.88 | 0.7 | 0.89 | 0.69 | 0.91 | 0.82 | 0.99 |
| Eclipse PDE UI 3.4.1 | 0.74 | 0.91 | 0.82 | 0.98 | 0.65 | 0.95 | 0.76 | 0.88 | 0.67 | 0.95 | 0.78 | 0.93 |
| Mylyn 3.1 | 0.74 | 0.87 | 0.81 | 0.99 | 0.7 | 0.9 | 0.73 | 0.87 | 0.71 | 0.92 | 0.79 | 0.88 |

**Table 13  AUC in experiment 1 and 2.**

| Dataset | LR | | NB | | DT | | MLP | | RF | | SVM | |
|---|---|---|---|---|---|---|---|---|---|---|---|---|
| | Exp1 | Exp2 | Exp1 | Exp2 | Exp1 | Exp2 | Exp1 | Exp2 | Exp1 | Exp2 | Exp1 | Exp2 |
| Apache Lucene 2.4 | 0.79 | 0.94 | 0.71 | 0.85 | 0.66 | 0.8 | 0.77 | 0.96 | 0.74 | 0.81 | 0.82 | 0.99 |
| Eclipse equinox framework 3.4 | 0.74 | 0.94 | 0.67 | 0.83 | 0.68 | 0.93 | 0.69 | 0.84 | 0.71 | 0.94 | 0.79 | 0.97 |
| Eclipse JDT Core 3.4 | 0.74 | 0.88 | 0.7 | 0.88 | 0.73 | 0.9 | 0.73 | 0.86 | 0.76 | 0.91 | 0.81 | 0.95 |
| Eclipse PDE UI 3.4.1 | 0.77 | 0.84 | 0.81 | 0.96 | 0.61 | 0.86 | 0.77 | 0.87 | 0.66 | 0.86 | 0.73 | 0.88 |
| Mylyn 3.1 | 0.68 | 0.86 | 0.79 | 0.99 | 0.57 | 0.82 | 0.8 | 0.96 | 0.67 | 0.85 | 0.68 | 0.87 |

Eventually, it has been observed that the results have been improved significantly by inducing Decomposed Halstead base metric into the conventional metrics.

## THREATS TO VALIDITY

The results of our experiment allow us to associate Decomposed Halstead base metrics with SFP. Nevertheless, before we could accept the result, we would have to consider possible threats to its validity.

### Internal validity

Internal validity refers to the potential risks to the data used in the experiment (*Jimenez-Buedo & Miller, 2010*). Concerning the size of the projects, sufficient comprehensible project size is taken. The projects of a very large size or very small size were ignored. The reason was the unavailability of either project's source code or fault information. Therefore, such very large size or small size projects may differ in the results reported.

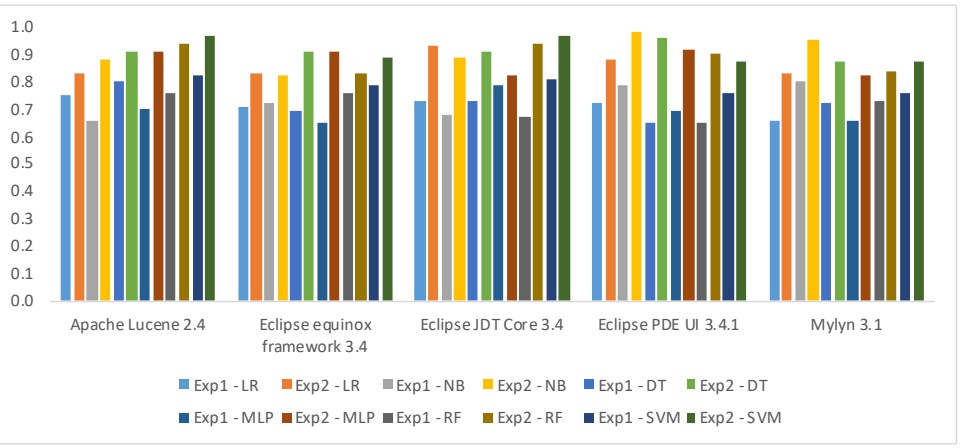

**Figure 6** Difference in accuracy by introducing decomposed Halstead base metric.

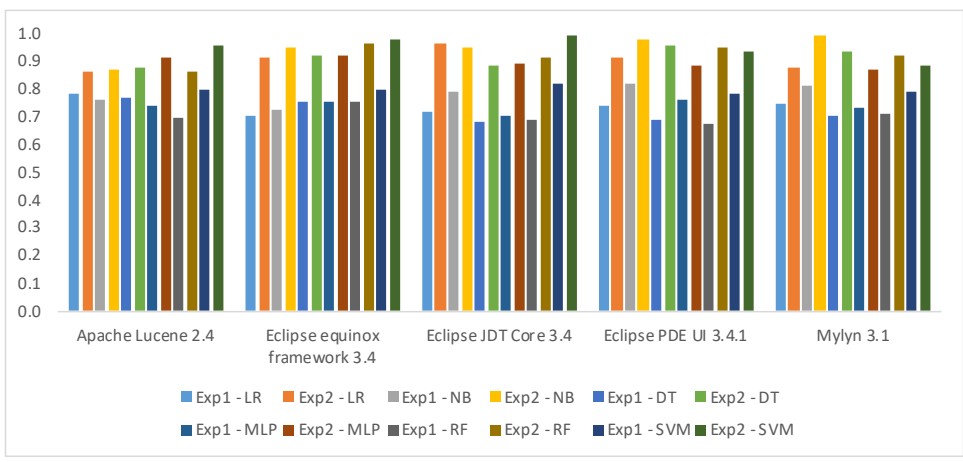

**Figure 7** Difference in F-measure by introducing decomposed Halstead base metric.

## External validity

External validity refers to the extent to which research findings can be generalized beyond the specific context of the study (*Jimenez-Buedo & Miller, 2010*). Regarding our experiments, the selected open-source projects are developed in Java, which sufficiently justifies the objective of the experiment and successfully demonstrates the experimental methodology. However, since the Decomposed Halstead base metric varies in different programming languages. The results may vary when using projects developed in languages other than Java.

## Construct validity

Construct validity implies the degree to which a measurement or assessment accurately captures and represents the underlying theoretical concept or construct it is intended to measure (*O'Leary-Kelly & Vokurka, 1998*). In our experiments, we include the coverage of

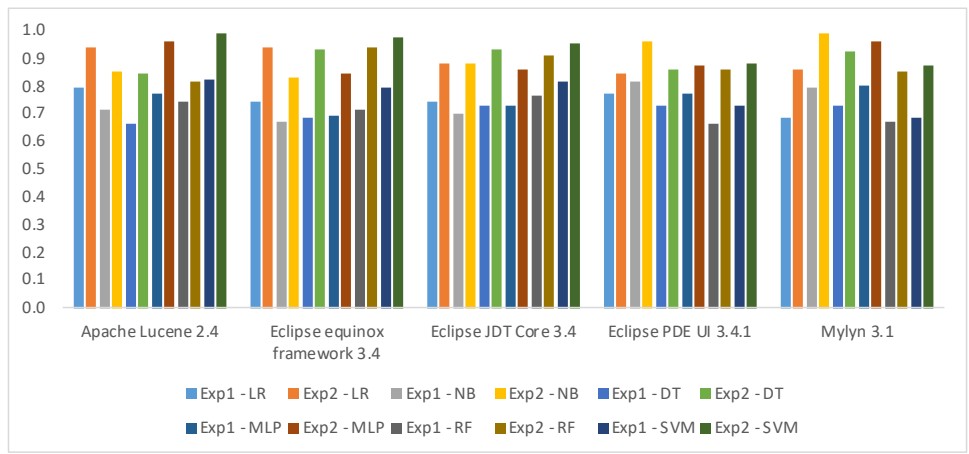

**Figure 8**  Difference in AUC by introducing decomposed Halstead base metric.

decomposed operators in our experiments, we may missed some builtin functions for such arithmetic operations. If we do so, the results will even be more promising. Hence, the impact of decomposed operators in SFP remains unrevealed.

## Conclusion validity

Conclusion validity refers to the certainty and accuracy of the inferences and conclusions drawn from the results (*García-Pérez, 2012*). The effectiveness of the decomposed Halstead base metric was evaluated based on the performance measures recorded in our study. The results we obtained are specifically related to the selected performance measures, namely Accuracy, F-measure, and AUC. The interpretations and explanations we provided are dependent on the defect labels assigned to the datasets we selected for our experiments. It is important to note that the results may vary when performing experiments on different datasets with varying distributions. Different datasets may exhibit different characteristics and defect distributions, which can influence the performance of the decomposed Halstead base metric. Therefore, caution should be exercised when generalizing the findings of our study to other datasets or scenarios.

## CONCLUSION AND FUTURE WORK

In this study, we conducted two experiments aimed at evaluating the performance of decomposed Halstead base metrics. The first experiment involved the utilization of the Halstead metrics suite in combination with McCabe, and LoC, as predictors. In the second experiment, we employed decomposed Halstead base metrics as predictors along with the Halstead derived, McCabe, and LoC metrics. The results have been reported in terms of Accuracy, F-measure, and AUC. The results obtained in Experiment-2 exhibited superior performance compared to the results reported in Experiment-1. Specifically, in terms of Accuracy, the maximum achieved value for the Lucene dataset was 0.97 when including the Decomposed Halstead base metric. Regarding F-measure, the maximum achieved value for Apache Lucene was 0.96. Furthermore, the maximum achieved AUC value was 0.98

for Lucene when considering the AUC metric. This clearly demonstrates the enhanced fault prediction capabilities achieved through the utilization of decomposed Halstead base metrics. This study focuses on the utilization of decomposed Halstead base metrics; however, further decomposing Halstead operands, variables and constants on behalf of their data types, can prove to be a prudent choice for exploitation. Additionally, apart from SFP, the Decomposed Halstead base metric can be applied in various other areas, including design patterns, impact analysis, software quality assessment, maintenance cost evaluation, productivity analysis, software vulnerability identification, changeability assessment, and more. Finally, it is worth noting that there are numerous other metrics that can also benefit from decomposition to enhance their efficiency. For instance, Cyclomatic complexity can be decomposed based on conditionals and decisions, while the Lines of Code metric can be decomposed into the number of literals, among other possibilities.

### Funding
The authors received no funding for this work.

### Competing Interests
The authors declare there are no competing interests.

### Author Contributions
- Bilal Khan conceived and designed the experiments, performed the experiments, performed the computation work, prepared figures and/or tables, and approved the final draft.
- Aamer Nadeem conceived and designed the experiments, analyzed the data, authored or reviewed drafts of the article, and approved the final draft.

### Data Availability
The data are available at Zenodo: Haonan Tong. (2019). AEEEM-JIRA-PROMISE (v1.0) [Data set]. Zenodo. https://doi.org/10.5281/zenodo.3362613.

### Supplemental Information
Supplemental information for this article can be found online at http://dx.doi.org/10.7717/peerj-cs.1647#supplemental-information.

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
