# Peer review of "Evaluating the effectiveness of decomposed Halstead Metrics in software fault prediction"

_PeerJ Computer Science, doi:10.7717/peerj-cs.1647_

## Round 0.1 · original submission · Major Revisions

As pointed out by reviewer 2 you need to improve your paper starting from:
- The writing of the paper is sub-par and it is very difficult to read through it
- Very shallow discussions and conclusions
- The methodology and experiment description are superficial and difficult to follow

Please address carefully all the comments provided by the two reviewers.

Reviewer 1 ·

Basic reporting

- Overall the article reports interesting results on using decomposed Halstead metrics to predict faults in source-code.

- Below are the comments for the different sections:

- Abstract:
- I would add an overview of the results by providing ranges of the obtained performance measurements.

- Literature review:
- I think "related work" is a more appropriate naming for this section as you are not really conducting a literature review in this paper (See. Literature review approach in Kitchenham, Barbara, et al. "Systematic literature reviews in software engineering–a systematic literature review." Information and software technology 51.1 (2009): 7-15.).
- Without a systematic approach, it is difficult to ensure good coverage of the literature. Another approach would be to conduct a system tertiary review on existing literature reviews on the topic (see Kitchenham 2009, cited above). This way, you can be sure that at least you are providing a good overview of the studies identified in the existing literature reviews. I would recommend going in this direction for better coverage of the literature in a systematic manner. My assumption is that the number of literature reviews on fault prediction is rather small and thus it will be relatively easy to build an overview systematically.

- Decomposition of Halstead operands:
- Did you try further decomposing the variables and the constants according to their data type? would this improve the results?

- Methodology:
- "Few conventional SFP metrics" -> "What metrics exactly?" I would mention McCabe and LoC here too (Page 12)

- Experimentation:
- Why McCabe, LoC are also used in the training of the machine learning models? did you try without? do they improve the performance? the Decision tree classifier can tell about the weight of these two measures in the overall prediction.
- In the cross-validation, I recommend leaving one dataset out for each validation round to show the performance of your model in datasets that it has not seen in the training
- The authors should document how balanced are the two classes in the datasets
- Ml Modeling: missing references for multilayer perceptron, random forests and evaluation measure

- Results:
- Since you are using decision trees, It would be very nice to provide the weights of the different features in the prediction to show the most predictive ones
- In the paragraph comparing the improvements between Experiment 1 and 2, it would be nice to incorporate the numbers about the minimum and maximum differences in the text

- Threats to validity:
- I suggest organizing and describing the threats to validity according to the categorization proposed in Wohlin, Claes, et al. Experimentation in software engineering. Springer Science & Business Media, 2012. I.e., internal validity, external validity, construct validity, conclusion validity

- Conclusion:
- Considering the reported findings, the future work suggestions seem promising
- The conclusion is quite short. I suggest providing a good summary of the paper and expend the future work part.
* * *
- General:
- The paper needs to be proof-read to fix the typos and improve the language
- Missing references to sections in several places
- There is a missing space (" ") in several citations
- For better clarity, Tables 6,7 and 8 should be provided in a tabular format and not as figures
- Please check that all abbreviations are defined in their first occurrence in the text

Experimental design

- The aim of the paper is clear but it would better to add concrete research questions
- The method is sufficiently well described

Validity of the findings

- The results are nicely presented and convincing.
- See the suggestion for threats to validity under the section "Threats to validity"

Reviewer 2 ·

Basic reporting

General comments:

The manuscript describes the application of decomposition to Halstead Metrics for the utilization in ML algorithms for Software Fault Prediction. The authors describe a case study where the decomposed Halstead metrics are used for SFP and their performance compared with normal Halstead Metrics. The authors report a consistent increase in the measured metrics and therefore the application of decomposition to Halstead metrics seems a valuable aid for ML-driven SFP.

However, the paper has many shortcomings that I summarize in the following:
- The writing of the paper is sub-par and it is very difficult to read through it
- Very shallow discussions and conclusions
- The methodology and experiment description are superficial and difficult to follow




Introduction:

line 56: the authors should provide brief definition of all methodologies instead of just listing them.

line 62: I do not understand the concept of "requirements" of ML-based SFP. The whole introduction in general is quite obscure and the reader is not able to understand the main concepts about how ML can be applied to SFP, which is vital for the understanding of the remainder of the paper.

line 95: Feature decomposition is not described nor motivated by the authors. The authors should describe clearly what feature decomposition is, which benefits are expected from the application of the technique, and how it has been applied previously in ML-based techniques. Otherwise, it is difficult to understand why the authors decided to apply decomposition to Halstead metrics in the first place.



Literature review:

How was the literature review performed? The authors mention two existing systematic literature reviews but I do not understand if all the papers mentioned below are extracted from these reviews.

line 115: six most applied classification methods in software defect prediction are: -> is this result taken from the mentioned literature reviews? Authors should explicitly say that if so. Also, if the items are not described it is not necessary to put them in an enumeration.

line 172: Halstead metrics are used in combination with McCabe and LoC (but not only) in the studies analyzed. Does this apply to all usages of Halstead metrics? Is this statement relevant for the study that has been performed?



Decomposition of Halstead operators

Some aspects of the decomposition procedure are not clear. The authors mention (line 209) that a base metric A will be decomposed into a1 ... an so that the sum of the value of metric A is equal to the sum of all the values of decomposed metrics. However, below the authors provide an example decomposition for the Java language, where many operators belong to both assignment and arithmetic operators (e.g., += and -=) and are therefore placed in both categories (assignment operators and arithmetic operators). For the base metric A, however, this will be a single operator. Doesn't this break up the constraint that A will be the sum of all the decomposed metrics? The authors should clarify this properly. A running example over a code snippet would be useful.



Methodology

line 280: what is an "indigenous" parser?

Some aspects of the methodology are vaguely described and not comprehensible. I especially refer to the "ML Algorithm" section.



Experimentation:

line 310: the authors selected five different software projects that have both source code and fault information. Were they the only 5 with such information? Or were they selected based to some criterion?

line 335: what is a Java-centric view of a project?


Results and discussion:

Instead of showing the difference in the bar graphs, it would be better to compare inside bar graphs the results for experiment 1 and experiment 2.

It looks like the results were consistently better for the second experimentation. The authors should better discuss this aspect and a rationale for such behaviour of ML algorithms


Threats to validity:

line 456: what is a "sufficient comprehensible project size"? The unavailability of fault information or project's source code is not a motivation for not considering very large projects.

In the threats to validity, the authors should discuss possible issues about the way the metrics were decomposed and speculate whether there are better possibilities for the splitting of Halstead metrics than the ones used.


Conclusion:

The discussion and conclusion sections of the manuscript are very limited in size and do not provide any hint on how researchers and practitioners can use the reported results.



Presentation and language

The manuscript would benefit significantly of a grammar check since many sentences are unclear
some examples
line 22: Halstead suite constitutes metrics related to operators and operands -> rephrase (the word constitutes is used in a consistely wrong way throughout the whole manuscript)
line 63: The discrete can be binary in nature like presence or absence of faults -> rephrase
line 83: The method-level metrics capture the method-level aspects -> redundant
line 96: "features are break down" -> broken
line 113: The systematic literature reviews (SLR) Catal and Diri (2009); Malhotra (2015) conclude the usage of public and private dataset -> unclear
line 192: "like, increment/decrement" -> substitute "like" with "For instance," or "e.g., " or similar.
line 308: "The datasets used in the literature does not contain information " -> do not
line 320: "The graphical representation of our experiments are shown in" -> is shown
line 383: classifier
line 462: However, since the Decomposed Halstead metric vary in different programming languages. The results may vary when using projects developed in languages other than Java. -> use a comma instead of a stop


All the citations should be harmonized with the text throughout the manuscript (e.g. in the first page: "it is found that faults are unevenly distributed in the software systemSherer (1995)" without any space or preposition)

All the section numbers are missing in the section references (e.g. line 103)

Is table 2 with the description and names of all Java operators really necessary?

There are too many acronyms used throughout the paper which make reading it particularly difficult.

The text in table 8 is cut and not readable

Figure 4: Bar charts or linear charts in general are preferrable to pie charts and 3d charts.

Experimental design

see above

Validity of the findings

see above

---

## Round 0.2 · Minor Revisions

Please address reviewer 1 comments and then we can proceed with the acceptance of your paper.

Reviewer 1 ·

Basic reporting

The quality of the manuscript has improved and most comments were addressed. There are, however, a few points that need to be further fixed/improved before publishing the article.

• "The summary of the results has been presented by furnishing ranges of achieved Accuracy, F-measure, and AUC across all experiments." this is not reflected in the current abstract
- I could not find this information in the paper abstract. Please check!

• In the introduction "A comprehensive discussion of software metrics is available in Ghani (2014); Fenton and Bieman (2014)."
- A more recent discussion of software metrics can be found at: "Abbad-Andaloussi, Amine. "On the relationship between source-code metrics and cognitive load: A systematic tertiary review." Journal of Systems and Software 198 (2023): 111619."

• In the introduction "By parsing the source code, we were able to extract the decomposed Halstead metrics from
124 these datasets". Here, it is important to provide a reference for the tool you use to extract these metrics

• Table 10 is roughly discussed in the findings and discussion section. I would suggest delving further into the discussion of the insights extracted from this Table.

• In "THREATS TO VALIDITY", short definitions of the different threats categories and references to that definitions should be provided, before introducing the associated threats and explaining how they relate to the defined threats.

• The abstract has a few typos:

- "In this article, we propose to decompose Halstead base metrics and evaluate **its** fault prediction capability." -> their
- ** ML classifier used included ..." -> the ML classier
- The performance of ** ML classifiers -> of the ML classifier

• In the introduction an empty space is missing before the reference in line 46.

• "CONCLUSION AND FUTUREWORK" -> Future work should be separated

Experimental design

- OK

Validity of the findings

- OK

Reviewer 2 ·

Basic reporting

The authors have addressed most of the comments provided by the reviewers.

The readability and the general writing of the manuscripts have improved.

I suggest acceptance of the manuscript.

As a minor aspect, I suggest to perform some fixing on figure 5: there are unnecessary borders in the graph, and I also suggest to avoid using pie charts in favor of a bar plot.

Experimental design

-

Validity of the findings

-

Additional comments

-

---

## Round 0.3 · accepted · Accept

Thank you for the modifications.